# Mathematically Modeling the Lexicon Entropy of Emergent Language

## Abstract

We formulate a stochastic process, FiLex, as a mathematical model of lexicon entropy in deep learning-based emergent language systems. Defining a model mathematically allows it to generate clear predictions which can be directly and decisively tested. We empirically verify across four different environments that FiLex predicts the correct correlation between hyperparameters (training steps, lexicon size, learning rate, rollout buffer size, and Gumbel-Softmax temperature) and the emergent language's entropy in 20 out of 20 environment-hyperparameter combinations. Furthermore, our experiments reveal that different environments show diverse relationships between their hyperparameters and entropy which demonstrates the need for a model which can make well-defined predictions at a precise level of granularity.

## 1 Introduction

The methods of deep learning-based emergent language provide a uniquely powerful way to study the nature of language and language change. In addressing these topics, some papers hypothesize general principles describing emergent language. For example, Resnick et al. [2020] hypothesize a predictable relationship exists between compositionality and neural network capacity, and Kharitonov et al. [2020] hypothesize a general entropy minimization pressure in deep learning-based emergent language. In many cases, these hypotheses are derived from intuitions and stated in natural language; this can lead to ambiguous interpretation, inadequate experiments, and *ad hoc* explanations. To this end, we study a general principle of emergent language by proposing a mathematical model which generates a testable hypothesis which can be directly evaluated through the empirical studies, akin to what we find prototypically in natural science.

We formulate a stochastic process, FiLex, as a mathematical model of lexicon entropy in deep learning-based emergent language systems[1] (ELS). We empirically verify across four different environments that FiLex predicts the correct correlation between hyperparameters (training steps, lexicon size, learning rate, rollout buffer size, and Gumbel-Softmax temperature) and the emergent language's entropy in 20 out of 20 environment-hyperparameter combinations.

There are three primary reasons for using an explicitly defined model for studying a topic like emergent language: clarity, testability, and extensibility. A mathematical model yields a *clear*, unambiguous interpretation since its components have precise meanings; this is especially important when conveying such concepts in writing. It is easier to *test* a model than a hypothesis articulated in natural language because the model yields clear predictions which can be shown to be accurate or inaccurate; as a result, models can also be directly compared to one another. Our experiments

---

[1] *Emergent language system* or ELS refers to the combination of agents (neural networks), the environment, and the training procedure used as part of an emergent language experiment.

Submitted to 36th Conference on Neural Information Processing Systems (NeurIPS 2022). Do not distribute.

reveal that different environments show diverse relationships between their hyperparameters and entropy which demonstrates the need for such clarity in making well-defined predictions at a precise level of granularity. Finally, mathematical models hypothesize a *mechanism* for an observed effect and not simply the effect itself (with a possibly *ad hoc* explanation). This is what facilitates their *extensibility* since a multitude of hypotheses can be derived from these mechanisms; furthermore, this "mechanical" nature allows future work to build directly on top of the model.

As mathematical models are seldom used to their full potential in studying emergent language, this paper is meant to serve as a reference and starting point for entire methodology of developing and testing such models. We articulate our contributions as follows:

- Defining a mathematical model of lexicon entropy in emergent language systems which we demonstrate to be accurate in predicting hyperparameter-entropy correlations.

- Presenting a case study of defining and empirically evaluating a mathematical model in emergent language.

- Provide a direct, intuitive comparison of the effects of hyperparameters on lexicon entropy across different environments.

We briefly discuss related work in Section 2. In Section 3, we introduce the mathematical model, FILEX, as well as the ELSs. Empirical evaluation is presented in Section 4 and discussed in Section 5, concluding with Section 6. Code is available at `https://example.com/reponame` (in supplemental material while under review).

## 2 Related work

For a survey of deep learning-based emergent language work, please see Lazaridou and Baroni [2020]. Contemporary deep learning-based emergent language research often aims at establishing and refining general principles about emergent language. In large part, these principles can be expressed as relationships between certain characteristics of the environment or agents (e.g., model capacity [Resnick et al., 2020], population size [Rita et al., 2022]) and properties of the emergent language (e.g., compositionality [Resnick et al., 2020, Rodríguez Luna et al., 2020], entropy [Kharitonov et al., 2020, Chaabouni et al., 2021, Rita et al., 2022], and generalizability [Chaabouni et al., 2020, Guo et al., 2021, Słowik et al., 2020]). Some of these works [Kharitonov et al., 2020, Khomtchouk and Sudhakaran, 2018, Resnick et al., 2020] make use of mathematical models to describe parts of the hypotheses and/or experiments, but these fall short of establishing a clear model which generates a testable hypothesis which is then evaluated through the empirical studies.

Pre-deep learning emergent language research frequently relied on mathematical models [Skyrms, 2010, Kirby et al., 2015, Brighton et al., 2005], but such models played a different role. Whereas these models were meant to account for some property of language observed in human language, the model presented in this paper is accounting for emergent language directly (and human language only indirectly). Thus, this paper presents a (mathematical) model of a (computational) model which, in the future, will be used to more directly study human language.

## 3 Methods

### 3.1 Model

FILEX ("fixed lexicon stochastic process") is a mathematical model developed from the Chinese restaurant process [Blei, 2007, Aldous, 1985], a stochastic process where each element in the sequence is a stochastic distribution over the positive integers (i.e., a distribution over distributions). The analogy for the Chinese restaurant process is a restaurant with tables indexed by the natural numbers; as each customer walks in, they sit at a random table with a probability proportional to the number of people already at that table. The key property here is that the process is *self-reinforcing*; tables with many people are likely to get even more. By analogy to language, the more a word is used the more likely it is to continue to be used. For example, speakers may develop a cognitive preference for it, or it gets passed along to subsequent generations as a higher rate [Francis et al., 2021].

**Algorithm 1** FⅠLEX pseudocode

```
1   alpha: float > 0
2   beta: int > 0
3   N: int > 0
4   S: int > 0
5
6   weights = array(size=S)
7   weights.fill(1 / S)
8   for _ in range(N):
9     W += sample_multinomial(W / sum(W), beta) / beta
10    w_copy = weights.copy()
11    for _ in range(beta):  # equivalent to normalized multinomial
12      i = sample_categorical(w_copy / sum(w_copy))
13      weights[i] += alpha / beta
14  return weights / sum(weights)
```

**Formulation** FⅠLEX is defined as a sequence of stochastic vectors indexed by $N \in \mathbb{N}^+$ given by:

$$\text{FⅠLEX}(\alpha, \beta, S, N) = \frac{\boldsymbol{w}^{(N)}}{\|\boldsymbol{w}^{(N)}\|_1} \tag{1}$$

$$\boldsymbol{w}^{(n+1)} = \boldsymbol{w}^{(n)} + \alpha \frac{\boldsymbol{x}^{(n)}}{\beta} \tag{2}$$

$$\boldsymbol{x}^{(n)} \sim \text{Multi}\left(\beta, \frac{\boldsymbol{w}^{(n)}}{\|\boldsymbol{w}^{(n)}\|_1}\right) \tag{3}$$

$$\boldsymbol{w}^{(1)} = \frac{1}{S} \cdot (1, 1, \ldots, 1) \in \mathbb{R}^S \tag{4}$$

where $\boldsymbol{w}^{(n)}$ is a vector of weights, $\alpha \in \mathbb{R}_{>0}$ controls the weight update magnitude, $\beta \in \mathbb{N}^+$ controls the variance of the updates, $S \in \mathbb{N}^+$ is the size of the weight vector (i.e., lexicon), and $\text{Multi}(k, \boldsymbol{p})$ is a $k$-trial multinomial distribution with probabilities $\boldsymbol{p} \in \mathbb{R}^S$. The pseudocode describing FⅠLEX is given in Algorithm 1. Conceptually, the process starts with an $S$-element array of weights initialized to $1/S$. At each iteration we draw from a $\beta$-trial multinomial distribution parameterized by the normalized weights.[2] This multinomial sample is multiplied by $\alpha/\beta$ and added to the weights so that the update magnitude is $\alpha$. This proceeds $N$ times. Since the sequence elements are the *normalized* weights, the elements are themselves probability distributions; thus, FⅠLEX is technically a sequence of distributions over distributions.

The two key differences between FⅠLEX and the Chinese restaurant process are the hyperparameters $S$ and $\beta$.[3] FⅠLEX has a fixed number of parameters so as to match the fact that the agents in the ELS have a fixed-size bottleneck layer, that is, a fixed lexicon. Secondly, $\beta$ is introduced to modulate the smoothness of parameter updates. It is closely connected to the fact that certain RL algorithms like PPO accumulate a buffer of data points from the environment with the same parameters before performing gradient descent.

## 3.2 Environments

To evaluate , we use four different reinforcement learning environments in our experiments. These are inhabited by two deep learning-based agents: (1) a sender agent which receives an observation and produces a message and (2) a receiver agent which receives a message (and possibly additional observation) and takes an action. The agent architecture and optimization are detailed Section 3.3.

**NODYN** The "no dynamics" environment is a proof-of-concept environment which is not intended to be realistic but rather to match as closely as possible the simplifying assumptions which FⅠLEX

---

[2]The $\beta$-trial multinomial sample is written as $\beta$ i.i.d. samples from a categorical distribution to draw parallels to PPO in Algorithm 2.

[3]Note that $\alpha$ in FⅠLEX is actually equivalent to the *inverse* of $\alpha$ in the Chinese restaurant process.

makes while keeping the same neural architecture in the environments below. As the name suggests, the primary simplification in this environment is that there are trivial dynamics, that is, every episode immediately ends with reward of 1 no matter what the sender or receiver do. The sender input and receiver output are identical to those of NAV, defined below. Just as FILEX assumes that every instance of word use is reinforced, this process reinforces every message which the sender produces.

**RECON**  The reconstruction game [Chaabouni et al., 2020], in the general case, mimics a discrete autoencoder: the input value is translated into a discrete message by the sender, and the receiver tries to output the original input based on the message. For a given episode, the sender observes $x \sim \mathcal{U}(-1, 1)$ and produces a message; the receiver's action is a real number $\hat{x}$, yielding a reward $(x - \hat{x})^2$.

**SIG**  The signaling game environment comes from Lewis [1970] and has been frequently used in the literature [Lazaridou et al., 2017, Bouchacourt and Baroni, 2018]. In this setup, the data is partitioned into a fixed number of discrete classes. The sender observes a datum from one of the classes and produces a message; the receivers observes this message, the sender's datum, and data points from other classes (i.e., "distractors"). The reward for the environment is 1 if the receiver correctly identifies the sender's datum among the distractors and 0 otherwise.

To eliminate the potential confounding factors from using natural inputs (e.g., image embeddings [Lazaridou et al., 2017]), we use a synthetic dataset. For an $n$-dimensional signaling game, we have $2^n$ classes. Each class is represented by an isotropic multivariate normal distribution with mean $(\mu_1, \mu_2, \ldots, \mu_n)$ where $\mu_i \in \{-3, 3\}$. Observations of a given are samples of its corresponding distribution. For example, in the 2-dimensional game, the 4 classes would be represented by the distributions: $\mathcal{N}((-3, -3), I_2)$, $\mathcal{N}((3, -3), I_2)$, $\mathcal{N}((-3, 3), I_2)$, and $\mathcal{N}((3, 3), I_2)$ (we use a 5-dimensional signaling game for our experiments with 32 classes). The motivation for this setup is minimal need for feature extraction while still using real-valued, stochastic inputs.

**NAV**  For a multi-step environment, we use a 2-dimensional, obstacle-free navigation task. The sender agent observes the $(x, y)$ position of a receiver and produces a message; the receiver moves by producing an $(x, y)$ vector. For a given episode, the receiver is initialized uniformly at random within a circle and must navigate towards a smaller circular goal region at the center. The agents are rewarded for both reaching the goal and for moving towards the center. An illustration is provided in Appendix A. The receiver's location and action are continuous variables.

### 3.3  Agents

**Architecture**  Our architecture comprises two agents, conceptually speaking, but in practice, they are a single neural network. The sender and receiver are randomly initialized at the start of training, are trained together, and are tested together. The sender itself is a 2-layer perceptron with tanh activations. The sender's input is environment-dependent. The output of the second layer is passed to a Gumbel-Softmax bottleneck layer [Maddison et al., 2017, Jang et al., 2017] which enables learning a discrete, one-hot representation.[4] The activations of this layer can be thought of as the words forming the lexicon of the emergent language. Messages consist only of a single one-hot vector (word) passed from sender to receiver. At evaluation time, the bottleneck layer functions deterministically as an argmax layer, emitting one-hot vectors. The receiver is a 1-layer perceptron which takes the output of the Gumbel-Softmax layer as input. The receiver's output is environment-dependent. An illustration and precise specification are provided in Appendices A and B.

**Optimization**  Although only our NAV environment involves multi-step episodes, using a full reinforcement learning algorithm across all environments benefits comparability and extensibility in future work. Specifically, we use proximal policy optimization (PPO) [Schulman et al., 2017] paired with Adam [Kingma and Ba, 2015] to optimize the neural networks. PPO is widely used RL algorithm which selected primarily for its stability (e.g., training almost always converges, minimal hyperparameter tuning); attempts to train with "vanilla" advantage actor critic did not consistently

---

[4]Using a Gumbel-Softmax bottleneck layer allows for end-to-end backpropagation, making optimization faster and more consistent than using a backpropagation-free method like REINFORCE [Kharitonov et al., 2020, Williams, 1992]. Nevertheless, future work may want to use REINFORCE for its more realistic assumptions about communication.

**Algorithm 2** PPO pseudocode

```
1  n_updates: int >= 0
2  buffer_size: int > 0
3
4  for _ in range(n_updates):  # outer loop
5    rollout_buffer = []
6    for _ in range(buffer_size):  # inner loop
7        episode = run_episode(model, environment)
8        rollout_buffer.append(episode)
9    update_parameters(model, rollout_buffer)
```

converge. We use the PPO implementation of Stable Baselines 3 (MIT license) built on PyTorch (BSD license) [Raffin et al., 2019, Paszke et al., 2019].

One relevant characteristic of PPO and similar algorithms is that in their training they contain an inner and outer loop analogous to FILEX (Algorithm 1); this is illustrated in Algorithm 2. The (main) outer loop consists of two steps: the inner loop which populates a rollout buffer with "experience" from the environment and the updating of parameters based on that buffer. What is important to note is that the buffer is populated with data from the same model parameters, and it is not until after this that model parameters change.

### 3.4 Hypothesis

Here we state the hypothesis used to evaluate FILEX. The sign of hyperparameter-entropy correlation observed in FILEX will be the same as what we observe for a corresponding hyperparameter in the ELSs. We can state this more formally as: for each pair of corresponding hyperparameters $(h, h')$ in FILEX and an ELS respectively,

$$\text{sgn}(\text{corr}(D)) = \text{sgn}(\text{corr}(D')) \tag{5}$$

$$D = \{(x, H(\boldsymbol{y})) \mid x \in X_h, \; \boldsymbol{y} \sim \text{FILEX}_{h=x}\} \tag{6}$$

$$D' = \{(x, H(\boldsymbol{y})) \mid x \in X_{h'}, \; \boldsymbol{y} \sim \text{ELS}_{h'=x}\} \tag{7}$$

$$H(\boldsymbol{y}) = -\sum_{i=1}^{S} y_i \log_2 y_i \tag{8}$$

where $\text{corr}(\cdot)$ is the Kendall rank correlation coefficient ($\tau$) [Kendall, 1938], $\text{FILEX}_{h=x}$ is the distribution over frequency vectors yielded by the model for hyperparameter $h$ set to $x$ (assume likewise for $\text{ELS}_{h'=x}$), $H$ is Shannon entropy, and $X_h$ is the set of experimental values for hyperparameter $h$. A "sample" from an ELS consists of training the agents in the environment, and estimating word frequencies by collecting the sender's messages over a random sample of inputs. Accordingly, our null hypothesis is that FILEX does not meaningfully correspond to the ELSs, and thus the signs of correlation would be expected to match with a probability $0.5$.

We intentionally formulate our hypothesis at this level of granularity: equality of direction (sign) of correlation rather stronger claims such as raw correlation: $|\text{corr}(D) - \text{corr}(D')| < \epsilon$ or mean squared error: $1/|X| \cdot \sum_{x \in X} (D(x) - D'(x))^2$. We select this level of direction of correlation for a few reasons. The level of simplicity of FILEX compared to the ELSs means that the unaccounted for factors would make supporting stronger hypotheses too difficult; furthermore, even if the hypothesis were defended, it would be less widely applicable for the same reasons. Additionally, the current literature tends to speak of the general principles of emergent language at the level of "relationships" and "effects" rather than exact numeric approximations [Kharitonov et al., 2020, Resnick et al., 2020].

**Corresponding Hyperparameters**    A key component of the hypothesis is the correspondence of hyperparameters of the ELSs with those of FILEX. These correspondences are the foundation for applying reasoning about FILEX to the ELSs; accordingly, they also determine how the model will be empirically tested. We present five pairs of corresponding environment-agnostic hyperparameters in Table 1. Although environment-specific hyperparameters can easily correspond with those of FILEX we chose the agnostic for ease of experimentation and comparison.

Table 1: Corresponding hyperparameters in the ELSs and FILEX.

| ELS | FILEX |
|---|---|
| Time steps | $N$ |
| Lexicon size | $S$ |
| Learning rate | $\alpha$ |
| Buffer size | $\beta$ |
| Temperature | $\beta$ |

Table 2: Kendall's $\tau$'s for various configurations. All values have a significance of $p \leq 0.01$.

| Environment | Time Steps | Lexicon Size | Learning Rate | Buffer Size | Temperature |
|---|---|---|---|---|---|
| FILEX | $-0.53$ | $+0.67$ | $-0.87$ | $+0.93$ | $+0.93$ |
| NODYN | $-0.81$ | $+0.12$ | $-0.74$ | $+0.07$ | $+0.58$ |
| RECON | $-0.17$ | $+0.93$ | $-0.35$ | $+0.84$ | $+0.68$ |
| SIG | $-0.49$ | $+0.15$ | $-0.16$ | $+0.30$ | $+0.49$ |
| NAV | $-0.81$ | $+0.36$ | $-0.84$ | $+0.20$ | $+0.68$ |

To identify these correspondences, it is important to understand the intuitive similarities between the ELSs and FILEX. Firstly, the weights of FILEX correspond the learned likelihood with which a given bottleneck unit is used in the ELS; in turn, both of these correspond to the frequency with which a word is used in a language. Each iteration of FILEX's outer loop is analogous to a whole cycle in the ELS of simulating episodes in the environment, receiving the rewards, and performing gradient descent with respect to the rewards (compare Algorithms 1 and 2).

Based on this analogy, we can explain the corresponding hyperparameters as follows. $N$ corresponds the number of parameter updates taken throughout the course of training the ELS (i.e., the outer loop of PPO). $S$ corresponds the size of the bottleneck layer in the ELS. $\alpha$ corresponds to the learning rate (i.e., magnitude of parameter updates) in the ELS. The ELS has two analogs of $\beta$. First, $\beta$ corresponds to the rollout buffer size of PPO because both control the number of iterations of the inner loop of training where episodes are collected before updating the weights. Second, $\beta$, more generally, control how smooth the updates to FILEX's weights are which makes it analogous to the temperature of the Gumbel-Softmax distribution in the ELS since a higher temperature results in smoother updates to the bottleneck's parameters.

## 4 Experiments

Our experiments consist of comparing the correlation between the hyperparameters of FILEX and the ELSs and the Shannon entropy of lexicon at the end of training. The entropy for the ELSs is calculated based on the bottleneck unit (word) frequencies gathered by sampling from the sender's input distribution. To gather data for FILEX, we run a Rust implementation of a sampling algorithm. Each experiment consists of a logarithmic sweep of a hyperparameter plotted against the entropy yielded by those hyperparameters (see Appendix B for details).

Each point in the resulting scatter plots corresponds to an independent run of the model or ELS with the hyperparameter on the $x$-axis and entropy on the $y$-axis. The plots also include a Gaussian convolution of the data points (the solid line) to better illustrate the general trend of the data. The plots are presented in Figure 1 with the rank correlation coefficients in Table 2.

## 5 Discussion

### 5.1 Model evaluation

Looking at the signs of correlations shows that FILEX makes the correct prediction 20 out of 20 times. Given a simple one-sided binomial test, the empirical data rejects the null hypothesis at $p < 0.001$.

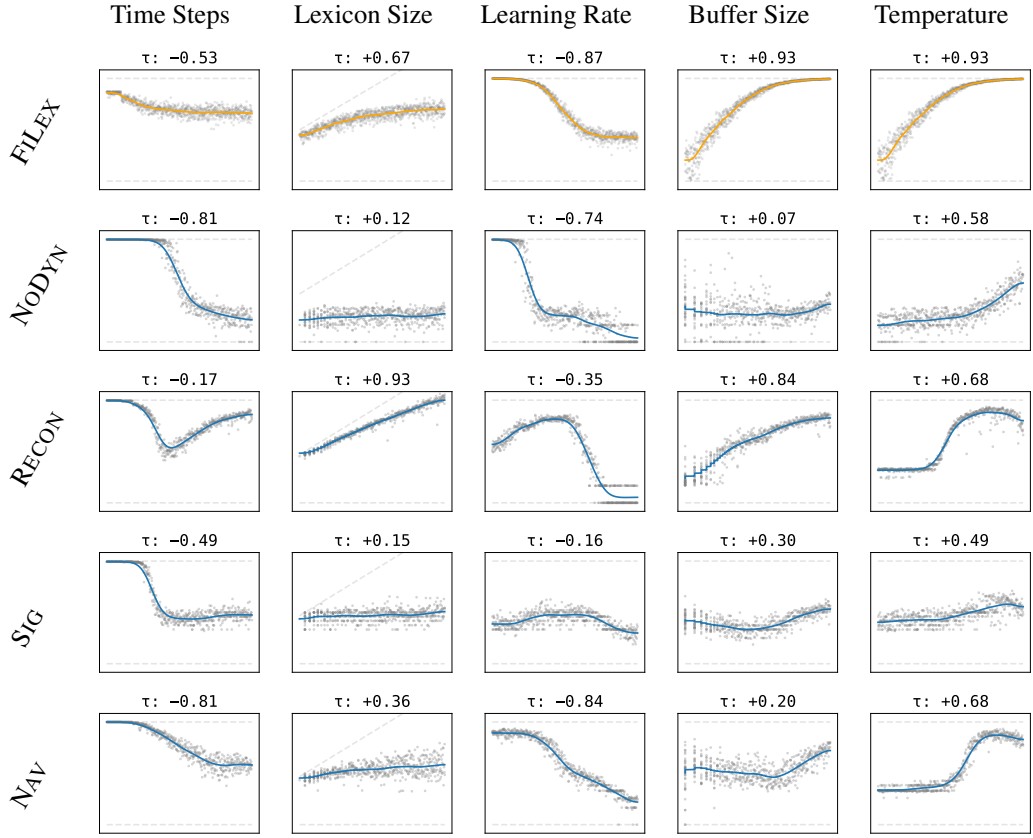

Figure 1: Plots of hyperparameters ($x$-axis, log scale) vs. entropy ($y$-axis) . Each row corresponds to a particular environment. Each column corresponds to a particular hyperparameter. All $y$-axes are on the same scale with the dashed lines representing min/max entropy. The points are individual runs and the lines are a Gaussian convolution of the points.

Although this number drops to 15 out of 20 if we require $|\tau| \geq 0.2$, the binomial test rejects the null hypothesis with $p = 0.02$ for this stronger hypothesis.

Though the directions of correlations predicted by FILEX are correct, looking at the plots show that ELSs do not always demonstrate the monotonicity predicted by the model. This is especially evident in *Time Steps* for RECON: moving left-to-right, the plot follows a similar path to the other environment and FILEX at first but then diverges halfway through with increasing entropy. A possible explanation of this is that RECON allows learning new, useful words more easily than SIG or NAV, meaning that additional training can lead to further improvement. The conclusion we draw from these plots is that FILEX correctly predicts a sort of baseline correlation between the hyperparameters and entropy. Other works, Kharitonov et al. [2020], Chaabouni et al. [2021] for example, find similar correlations between entropy and bottleneck temperature. Nevertheless, this correlation can be overridden by the specifics of the environment.

## 5.2 Environment variability

When looking beyond just the direction of correlation at the slopes and shapes of the curves, the four ELSs all present unique set of relationships between entropy their hyperparameters. This implies that none of these environments are reducible to each other, that is, we cannot make observations about one environment and automatically assume they apply to other environments. Certainly this makes an researcher's task harder as learning general principles would not be possible from a single environment. Furthermore, there is a sensitivity to hyperparameters *within* a given environment, which would imply that discovering general principles within single environment could not be done with just a single set of hyperparameters.

Although this diversity in behavior makes modeling it more difficult, it also shows the importance of precision we get from a mathematical model. For example, say RECON has not been empirically tested and we wanted to predict the lexicon size-entropy relationship in RECON. It is the case that we could simply observe the positive correlations in the other environments and predict the same RECON, but we could easily over-extrapolate and predict a relatively shallow slope when RECON's slope is relatively steep. What this paper's model, hypothesis, and evaluation offer in this situation is not a more detailed prediction but a "prepackaged" prediction which is precisely stated and supported by data.

## 5.3 Applications to future work

There are two primary ways in which FILEX can be applied in future research. First, the model can be applied to and tested against further phenomena in emergent language (i.e., it is *extensible*). The fact that it is formulated mathematically means that it does not just predict correlations but *mechanisms* which account for the correlations. For example, FILEX's $\beta$ hyperparameter was designed to account for *Buffer Size* and the *Temperature* experiment was conducted after the fact. The fact that FILEX describes both *Buffer Size* and *Temperature* with the same hyperparameter suggests that similar mechanisms account for their positive correlations with entropy. This statement about similar mechanisms, on the other hand, is not present set of one-off hypotheses about hyperparameter-entropy correlations derived from intuition. Second, FILEX and accompanying experiments provide an easy way for future research to discover confounding factors in their experiments. For example, an experiment might show that entropy decreases as rewards are scaled up, yet FILEX would suggest that this might be equivalent to simply increasing the learning rate rather than being its own unique cause of the effect on entropy.

## 5.4 Methodological difficulties

The greatest challenge in the methodology of this work is not the formulation of the model but rather evaluating the quality of the model. In part, this is on account of a lack of established baseline model—comparative analysis ("which is better?") is significantly easier than absolute analysis ("how good is this?") yet requires an adequate baseline to compare against. But more significantly, the granularity of experimentation is a design decision with no obvious answer.

For example, merely comparing the signs of rank correlations is very coarse-grained as it makes minimal assumptions about the data (e.g., linearity, absence of outliers) and captures very little information about the data. Naturally, it is easier to apply such an analysis, and as mentioned before, researcher typically phrase hypotheses in terms of such correlations, but it can only offer minimal support for applicability of the model to the actual system. On the other hand, evaluating the model's ability to predict exact behavior of the system (e.g., measuring mean squared error of the model's predictions) can establish a more precise link between model and system but might miss more general but important similarities. For example, *Lexicon Size* for FILEX and NAV might show similar trends, but be different by a constant, yielding a high mean squared error.

A subtle but significant methodological difficulty is the selection of hyperparameters. In RECON's *Time Steps* plot, it is easy to see that changing the range of hyperparameters could easily yield either a positive or a negative correlation when in reality there are both. To a certain extent, this can be resolved be choosing a "reasonable" range of hyperparameters based on values are typically, but this is of little help to selection of FILEX's hyperparameters as there is no "typical usage." For example, FILEX for $\beta = 1$ and $\beta = 100$ yield significantly different distributions, but there is no obvious *a priori* reason to say that one value of $\beta$ should be preferred over the other for comparing to the ELSs. Although additional hyperparameters increase the range of phenomena which the model can account for, the additional degrees of freedom can weaken the model's predictions by introducing confounding variables (cf. overparameterization).

One of the primary contributions of this work is to serve as a case study and example of working with explicitly defined models in studying deep learning-based emergent language. Thus, this paper is starting point for future work to improve upon. One of the most important improvements would be finding a more rigorous way to select "reasonable" experimental hyperparameters. Additionally, it would be better to develop the hypothesis and experimental in full before performing any evaluation; the process was somewhat iterative in this paper.

## 6 Conclusion

We have presented FILEX as a mathematical model of lexicon entropy in deep learning-based emergent language systems and demonstrated that, at the level of correlations, it accurately predicts the behavior of our emergent language environments. Opting for a mathematical model possesses the benefits of having a clear interpretation, making testable predictions, and being reused for new predictions in future studies. Although the model's hypothesis was testable, the process is not free from non-trivial design decisions which affect the quality of evaluation. Nevertheless, this paper serves as starting point and example of how more rigorous models can be applied to the study of emergent language.

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

## A  Emergent language system illustration

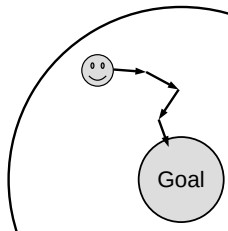

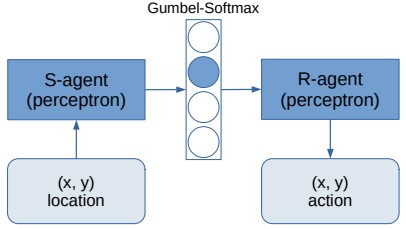

(a) The receiver (pictured) is rewarded for moving towards the goal region at the center in the NAV environment.

(b) The agent architecture for NAV.

Figure 2

## B  Experiment parameters

Each experiment uses a logarithmic sweep across hyperparameters; the sweep is defined by Equation 9, where $x$ and $y$ are the inclusive upper and lower bounds respectively and $n$ is the number steps to divide the interval into. The floor function is applied if the elements must be integers.

$$\text{LS}(x, y, n) = \left\{ x \cdot \left(\frac{y}{x}\right)^{\frac{i}{n-1}} \,\middle|\, i \in \{0, 1, \dots, n-1\} \right\} \tag{9}$$

| Hyperparameter | Default | Low | High | Steps |
|---|---|---|---|---|
| $N$ | $10^3$ | $10^0$ | $10^3$ | 1000 |
| $S$ | $2^6$ | $2^3$ | $2^8$ | 1000 |
| $\alpha$ | 1 | $10^{-3}$ | $10^3$ | 1000 |
| $\beta$ | 8 | $10^0$ | $10^3$ | 1000 |

Table 3: Hyperparameters for the empirical evaluation of FILEX. "Low" and "High" refer to the logarithmic sweep used for that experiment; default values used for all other experiments.

| Hyperparameter | Default | Low | High | Steps |
|---|---|---|---|---|
| Time steps | $2 \cdot 10^5$ | $10^2$ | $10^6$ | 600 |
| Bottleneck size | $2^6$ | $2^3$ | $2^8$ | 600 |
| Learning rate | $3 \cdot 10^{-3}$ | $10^{-4}$ | $10^{-1}$ | 600 |
| Buffer size | $2^8$ | $2^3$ | $2^{10}$ | 600 |
| Temperature | 1.5 | $10^{-1}$ | $10^1$ | 600 |

Table 4: Hyperparameters for the empirical evaluation of FILEX. "Low" and "High" refer to the logarithmic sweep used for that experiment; default values used for all other experiments. Please see code for further details and default values.

