# OpenReview forum: "Mathematically Modeling the Lexicon Entropy of Emergent Language"
_NeurIPS.cc/2022/Conference — NeurIPS 2022 Submitted_

### Official Review · Reviewer_YbMp · 2022-07-09

**Rating:** 3
**Confidence:** 4
**Soundness:** 1 poor
**Presentation:** 3 good
**Contribution:** 2 fair

**Summary:**

This paper fits into the Emergent Language literation and argues that prior work has been insufficiently rigorous when presenting their hypotheses. This work is presented as a starting point for mathematical methodology in developing and testing models of emergent language.

Their main contribution is an approach "FiLex" to understanding models of emergent language influenced by the Chinese Restaurant process (CRP). The relation here is to describe words as having a similar property to the tables in the CRP in that their usage is self-reinforcing. This approach is detailed in Section 3.1 with Algorithm 1 and the Formulation on page 3.

Their hypothesis, presented in Section 3.4, is that hyperparameters in FiLex correlate with hyperparameters commonly found in emergent language setups: time steps, lexicon size, learning rate, buffer size, and temperature. Their specific statement is that the sign of the correlation between those hyperparameters and entropy will be the same for FiLex as it is for the ELS setups.

They experimentally test this hypothesis by comparing FiLex against 4 ELS setups in Table 2 and Figure 1 and then make claims in Sec 5 (discussion) that this model lets practitioners more rigorously understand confounding factors.

**Questions:**

1. It's not clear in this paper whether we are asking whether the mathematical model that this process produces can answer questions of current ELS systems or are we asking whether it can help us illuminate where these systems are weak. Which one is it? The second is much more significant imo.

2. Why should the sign of the per-hyperparameter entropy correlation be predictive about what my approach will do? Is there some foundation for that in other literature?

3. What is the resulting word distribution of the CRP at steady state?

4. My suggestion is to return to the models you've made (the 4 methods NoDyn Recon Sig and Nav) and try and see what, if anything, is invariant and unit-independent amongst these. I think that will bear more interesting fruit than this approach starting w CRP.

**Limitations:**

Societally, this is fine. Wrt limitations, see the above S&W section.

**Strengths And Weaknesses:**

Originality:

This paper is original. It's taking a well-known idea (CRP) and applying it to a domain (emergent language) where the procedure is potentially beneficial for attaining understanding. It's unclear to me that this is actually the right approach for this domain and that's something that warrants greater discussion in the paper, but it's still an interesting direction that has merit.


Quality:

I don't think that this approach makes sense to do in ELS. I love the motivation to have better mathematical foundations in the space, but I disagree that this approach, or at least as presented, is the right path.

First, why should it be the case that methods with same sign correlations in individual hyperparameters are actually indicative of each other? This is much too weak of a statement to be predictive as we could manufacture this artificially without much issue. The argument given at the end of section 3 that then qualitatively associates the variables doesn't fix this issue; changing just the algorithm used from PPO to something else that doesn't have a buffer size shows how fragile that is.

The explanation in 3.4 (L173-L180) points to this as well, suggesting that there are too many factors unaccounted for (due to FiLex's simplicity) to do much else besides this. But that's exactly what the paper set out to do at the beginning after discussing how prior work skipped out on this important step. Perhaps they did so for this same reason?

Sec 5, the evaluation, talks about making predictions ("[... FiLex] makes the correct prediction 20 out of 20 times.") and how that wouldn't happen if it wasn't predictive. But then the graphs (and the discussion) shows how tenuous that is. It's not very helpful for my research to run this method a bunch of times and hope that the correlation remains positive versus just varying a hyperparameter in my actual model and seeing what that does. Perhaps the authors are thinking that I cannot run my actual model that much because, say, it's a real robot. Okay that's fair, but I have no idea if this method would actually work in that setting and little belief that my model in totality would really be correlated given that a handful of hyperparameters are.

Clarity:
The paper is clear enough. The one area which would have been more helpful is to coalesce Table 2 and Figure 1 to be more apparently connected. It takes longer than I would have liked to understand what was going on in them (which are the results) and the stories they told.

Significance:

This paper is only significant in the world where the simple model is predictive of much bigger and more interesting models. The ones examined are not that, nor is it even assessed as predictive of anything in those settings.

For it to get to a place where we can confidently say that this is interesting for real-world settings, it would need to be aligned with actual (emergent) language learning. That is unfair to evaluate wrt learning new words in this setting (as neither FiLex nor most ELS systems are doing that), but it is fair to ask whether the learned language distributions are the same. Is that true? Is FiLex learning similar word distributions to the ELS models? More to the point, because FiLex's learned distribution won't ever be different given its simplicity, is it Zipfian? We expect that to if it's ever going to be able to model a full language.

---

> ### Author Response · Authors · 2022-08-01
> **Response to YbMp**
>
> Thank you for your review.
> Below, we provide a brief response to your comments and questions.
>
> > [Question 1] It's not clear in this paper whether we are asking whether the mathematical model that this process produces can answer questions of current ELS systems or are we asking whether it can help us illuminate where these systems are weak. Which one is it? The second is much more significant imo.
>
> If I understand correctly, this question is asking whether the purpose of FiLex is to increase understanding of emergent language systems as they stand now or to diagnose problems while developing future emergent language systems.
> This paper is primarily doing the former: providing a mathematical model of a given system which is simple enough to grasp conceptually.
> Nevertheless, this combination of precision and conceptual simplicity would be invaluable in diagnosing and explaining undesirable behaviors while developing emergent language systems.
>
>
> > [Question 2] Why should the sign of the per-hyperparameter entropy correlation be predictive about what my approach will do? Is there some foundation for that in other literature?
>
> I believe this question is asking why the sign of the correlation between a given hyperparameter and entropy is indicative of how the emergent language system behaves as a whole.
> While entropy is one of many characteristics of an emergent language, it is a central concept in information theory and has receive attention from prior work [[Kharitonov et al., 2020](http://proceedings.mlr.press/v119/kharitonov20a.html); [Chaabouni et al., 2021](https://www.pnas.org/doi/full/10.1073/pnas.2016569118)].
> Thus, understanding the basic correlation (i.e., positive or negative) between entropy and the various hyperparameters of the emergent language system is a fundamental part of understanding an emergent language as a whole.
>
>
> > [Question 3] What is the resulting word distribution of the CRP at steady state?
>
> The CRP and FiLex are both distributions over distributions.
> So a single _sample_ from FiLex would be a distribution over the words in the vocabulary.
> A handful of illustrated samples (distributions) could be included in the appendix if that is being requested.
>
>
> > [Question 4] My suggestion is to return to the models you've made (the 4 methods NoDyn Recon Sig and Nav) and try and see what, if anything, is invariant and unit-independent amongst these. I think that will bear more interesting fruit than this approach starting w CRP.
>
> To my understanding, you are suggesting that we start the research process by looking at the commonalities among the environments and noting any consistent trends rather than starting with the observation that emergent language-learning is self reinforcing and expressing that in a model.
> To a certain extent, this is close to how the development of this paper went.
> It began with observing hyperparameter-entropy plots (e.g., learning rate-entropy) and observing the trends.
> When trying to explain the rends we observed, we found that it was clearest to express them in a mathematical model like FiLex.
> From there, we developed FiLex into its current form and verified its predictions on further environments.
>
> [Weakness - Quality] changing just the algorithm used from PPO to something else that doesn't have a buffer size shows how fragile that is.
>
> If we used an algorithm that did not have a buffer size, FiLex could still apply to that model.
> It would just be the case that there is one fewer correlation between FiLex and the emergent language system.
> We would still have the correlation between $N$ and time steps, $S$ and lexicon size, $\alpha$ and learning rate, and $\beta$ and Gumbel-Softmax bottleneck temperature.

---

### Official Review · Reviewer_uYiF · 2022-07-10

**Rating:** 3
**Confidence:** 3
**Soundness:** 2 fair
**Presentation:** 2 fair
**Contribution:** 3 good

**Summary:**

The paper proposes a stochastic process FILEX developed from the Chinese restaurant process, to mathematically model the lexicon entropy of emergent language between multiple agents (the speaker and the listener) in ELS (emergent language system) environments. The authors make correspondences between the FILEX and ELS to evaluate FILEX in real ELS environments. Experimental results on four ELS environments show that FILEX can correctly predict the correlation between hyperparameters and the lexicon entropy of well-trained emergent language in real ELS systems.

**Questions:**

1. Does the self-reinforcing assumption of emergent language in line 79 make sense in all ELS environments? Are there any evidences in the literature or experimental results?
2. What’s the idea or intuitions behind the hyperparameter-entropy correlation in line 162? Why can it be used to evaluate whether FILEX matches the real ELS system?


**Limitations:**

According to the authors, there are weaknesses in this work that have not been addressed. Details are as above.
This paper does not have any potential negative societal impact.


**Strengths And Weaknesses:**

In this paper, the authors propose a mathematical model of lexicon entropy of emergent language between agents. Moreover, the authors conduct extensive experiments to verify the model, and the results show its effectiveness.
Strengths:
1.	The paper provides an inspiring idea of mathematically modeling the properties of the emergent language without training in real environments. Besides, the mathematical description is more precise and testable than natural language.
2.	The experimental results demonstrate the effectiveness of the proposed FILEX, which verifies the feasibility of the idea.
Weaknesses:
1. The mathematical model may be not rigorous enough.
----a) The proposed FILEX is based on the assumption that word use is reinforced in emergent language in sec 3.1. However, the paper only provides evidence in human language but not emergent language between agents. The paper does not prove the assumption fits all ELS environments either.
----b) The correspondences of hyperparameters between FILEX and ELS in line 193 are based on analogy, thus the relationships between FILEX and ELS maybe not strong enough to support the alignments. Besides, beta in FILEX is corresponding to both buffer size and temperature in ELS, which may break the independence of the two hyperparameters despite the reason given in the paper.
2. The experiments seem to be not well-organized and insufficient somewhat.
----a) In sec 3.4, the meaning of the hyperparameter-entropy correlation and the reason why it can be used to evaluate whether FILEX matches ELS is not given.
----b) It is better to add the experimental evidence of the self-reinforcing assumption of emergent language in the four ELS environments, which is the basic assumption of the proposed FILEX.
----c) In line 173, as a testable mathematical model, FILEX is expected to provide more precise information, thus the equality of sign of correlation as the metric seems a little too weak. It would be better if the authors add some stronger metrics.
3. The organization and presentation of the paper could be improved.
----a) The introduction section misses necessary background introductions to the key concept “emergent language” and “lexicon entropy”, which may confuse readers from a broad domain.
----b) The paper lacks a formal definition of the problem or task. Besides, it is better to give the input, output, and goal of FILEX before the technical details in sec 3.1.
----c) In sec 3.4, it is better to provide more explanations about the obscure hyperparameter-entropy correlation, which may be unfamiliar to many readers.
----d) The paper may be not well-organized. For example, the environments in sec 3.2 and result analysis in sec 5.1 should be included in the experimental parts.
----e) There are some typos and small mistakes in the paper. For example, line 9 in Algorithm 1 is not correctly initialized and used elsewhere; “Section 3.3” -> “in Section 3.3” in line 102.

---

> ### Author Response · Authors · 2022-08-01
> **Response to uYiF**
>
> Thank you for your review.
> Below, we provide a brief response to your comments and questions.
>
> > [Question 1.1] Does the self-reinforcing assumption of emergent language in line 79 make sense in all ELS environments?
>
> Yes, generally speaking, emergent languages derived from reinforcement learning are self-reinforcing.
> This is because of the following cycle:
> - Used words are learned more; that is, when a word is used, the sender and receiver networks are adjusted along the gradient given by the reward signal.
> - Learned words are used more; that is, when the sender and receiver agree more on the meaning of a word, it will be used more because it is effective in maximizing the reward.
>
> This is somewhat like a multi-armed bandit problem where the expected value of one of the arms increases when it is pulled, which will increase its probability of being pulled again in the future.
> Naturally, the dynamics of each individual environment cause the learning process to diverge from idealizations like FiLex, but the underlying principle of self-reinforcement is still there.
>
> > [Question 1.2] Are there any evidences in the literature or experimental results?
>
> Nothing explicit that we know of aside from our own experiments.
> The Pitman-Yor process, a generalization of the Chinese restaurant process (both of which are self-reinforcing), produces power-law distributions similar to those we find in natural language, suggesting the natural language lexical distributions can be explained with a self-reinforcing model [[Teh, 2006]](https://aclanthology.org/P06-1124.pdf).
> Additionally, [Mordatch and Abbeel [2017]](https://ojs.aaai.org/index.php/AAAI/article/view/11492) use a Chinese restaurant process-based reward to inductively bias an emergent language towards compositionality.
>
> > [Question 2] What’s the idea or intuitions behind the hyperparameter-entropy correlation in line 162? Why can it be used to evaluate whether FILEX matches the real ELS system?
>
> The primary phenomenon which FiLex is intended to model is the interaction between ELS hyperparameters and resulting lexical distribution.
> The entropy of a discrete distribution is an intuitive one-number summary of the distributions shape; additionally, entropy is a central value in information theory and the theory of communication.
> Thus, looking at the hyperparameter-entropy correlation, we can answer the question, "Is FiLex a good way to conceptualize how ELS hyperparameters affect the lexicon entropy we see in the ELS?"
>
> > [Weakness 1.b] The correspondences of hyperparameters between FILEX and ELS in line 193 are based on analogy, thus the relationships between FILEX and ELS maybe not strong enough to support the alignments.
>
> This is an intended to be a feature and not a bug.
> When we work on complex systems like emergent language environments with deep neural networks, creating precise mathematical models is generally impractical, if not impossible.
> So instead we construct mental/conceptual model approximating how the complex system works; correspondences here, naturally, can only be analogous and not rigorously demonstrated.
> FiLex is meant to be a formalization of a conceptual model of a complex system rather than definitively proving something about a simpler system (e.g., valuation iteration is guaranteed to converge, PAC learning theorems).
>
> > [Weakness 2.c] In line 173, as a testable mathematical model, FILEX is expected to provide more precise information, thus the equality of sign of correlation as the metric seems a little too weak. It would be better if the authors add some stronger metrics.
>
> Given that most of the literature in the field only speaks of correlations (e.g., effect of Gumbel-Softmax temperature on communication entropy [[Kharitonov et al., 2020]](http://proceedings.mlr.press/v119/kharitonov20a.html)), this paper is making predictions at the same level of granularity.
> Additionally, in order to make more precise predictions (e.g., the slope of the line), the model become too complex to be reasoned about conceptually.

---

> > ### Comment · Reviewer_uYiF · 2022-08-09
> > **Thanks for the response**
> >
> > Thanks for the authors’ responses. This clears up some of my confusions about the intuitions and experiments. However, I still think that the current manuscript cannot overlook the weakness listed above, e.g. the experiments are not enough to support the intuitions, and the paper is not clarified clearly. The paper should be further improved from several perspectives.

---

### Official Review · Reviewer_4yzD · 2022-07-11

**Rating:** 6
**Confidence:** 4
**Soundness:** 3 good
**Presentation:** 3 good
**Contribution:** 3 good

**Summary:**


This paper proposes a stochastic process, FILEX, to abstract out the essence of deep learning based emergent language systems. FILEX follows the intuition that the more a word is used, the more it will be used in the future. On four experimental settings, the authors show that the correlation between parameters of FILEX and the lexicon entropy is similar to the correlation between hyper parameters of neural networks and the lexicon entropy.

**Questions:**

How would the conclusion generalize to setting when the vocabulary is large and when the setting is compositional?

**Limitations:**

The authors have properly discussed the limitations

**Strengths And Weaknesses:**

strength

- This work attempts to construct a simple enough theory for understanding emergent language. Such attempt is generally valuable for the overall community as the filed of emergent language itself is, emerging.
- The proposed method is simple enough to mimic the model behavior thus providing a level of abstraction/ simplification for understanding. Additionally, I believe similar observation is also discussed in the previous VAE literature, just for the authors information[1].
- This work is inspiring to me and I believe such work should also be inspiring to future work. Though I am not fully convinced with the existing experiments (see below), I am happy to see either future work may reinforce its conclusion or turn them over.

weakness

- The proposed methods is too simple to miss important details of many aspects of neural networks. When the authors link the parameters to of FILEX to the neural network hyper parameters, their reasons are more about intuition rather than rigorously mathematically discussions.
- The experimental settings are too simple and may be hard to generalize to more complicated setting. One direct generalization is whether the conclusion will hold if the vocabulary size is large (say 10K) and compositional (say a sentence of length 10).

[1] Alemi et. al. ICML 2018. Fixing a Broken ELBO

---

> ### Author Response · Authors · 2022-08-01
> **Response to 4yzD**
>
> Thank you for your review.
> Below, we provide a brief response to your comments and questions.
>
> > [Weakness] The proposed methods is too simple to miss important details of many aspects of neural networks. When the authors link the parameters to of FILEX to the neural network hyper parameters, their reasons are more about intuition rather than rigorously mathematically discussions.
>
> To a certain extent, this is an unsolvable problem in emergent language.
> Emergent language environments along with deep neural network agents are too complicated for direct proof of properties that are of interest.
> Instead, it is common practice to make intuitive hypotheses about emergent language.
> Thus, we are trying to introduce a more (though not fully) formal way of making a hypothesis that balances formality with ease-of-use and applicability.
>
>
> > [Question] How would the conclusion generalize to setting when the vocabulary is large and when the setting is compositional?
>
> Given the trends observed in our experiments and simplicity of the environments, we would expect that increasing the vocabulary beyond $256$ would continue the rends shown in Figure 1.
> If by compositional vocabulary it is meant that the messages consist of multiple symbols instead of one, we do not see any immediate indication that the results would be different.
> We wanted to establish FiLex for a simpler set of environments before moving to more complex ones.

---

> > ### Comment · Reviewer_4yzD · 2022-08-09
> > **Thank you for your clarification**
> >
> > I'll keep my initial rating.

---

### Author Response · Authors · 2022-08-01
**General comment**

Thank you to all of the reviewers and chairs for their time and effort.

We would like to respond to a general theme in the reviews regarding the framing of the paper.
Given an emergent language system built on top of (deep) neural networks and reinforcement learning (including but not limited to the ones presented in the paper), there are three levels at which we can "model" the system:
1. With a fully formalized model that takes into account all aspects of the emergent language system.
    This is the Python code which formally describes the exact process used to generate the emergent language.
2. With intuitions about how different hyperparameters of the emergent language system relate to the properties of the emergent language (e.g., entropy).
3. With a simplified mathematical model that formalizes some of the key intuitions about the emergent language system.

The issue with (1) is that although the full model is accurate, it is difficult to reason and hypothesize about directly.
The issue with (2) is that intuitions are often expressed qualitatively which limits their clarity and reusability; this is the way most papers in the field of emergent language explain their hypotheses and results.
(3), the approach we take with FiLex, is intended to combine some of the formality of (1) with the ease-of-use and intuitiveness of (2).
In light of this, the intended framing of the paper is that we are providing an example of a pragmatic formalization of aspects of emergent language research which are seldom formal at all.
In contrast, we are not trying to formally prove with FiLex that various hyperparameters and entropy are always related in a certain way in full emergent language systems.

---

### Meta-Review · Area_Chair_rG8a · 2022-08-29

**Recommendation:** Reject
**Confidence:** Certain

**Metareview:**

This paper proposes FiLex -- a mathematical model to capture lexicon entropy in emergent language systems. The paper tackles an important and interesting problem in a field (emergent language) where relatively less theory currently exists. However, the reviewers find the experiments not convincing enough (e.g. they do not evaluate actual emergent language and instead use human languages) and lacking in scale. I do think the paper has some merits and can be strengthened further by addressing the reviewer comments, but the current version unfortunately seems below bar for acceptance.

**Award:**

No

---

### Decision · Program_Chairs · 2022-09-14

Reject